# Illness-related suffering and need for palliative care in Rohingya refugees and caregivers in Bangladesh: A cross-sectional study

**Megan Doherty**[1,2,3]*, **Liam Power**[4], **Mila Petrova**[5], **Scott Gunn**[6], **Richard Powell**[7], **Rachel Coghlan**[8], **Liz Grant**[9], **Brett Sutton**[10], **Farzana Khan**[11]

1 Department of Pediatrics, Children's Hospital of Eastern Ontario, Ottawa, Ontario, Canada, 2 Department of Pediatrics, Faculty of Medicine, University of Ottawa, Ottawa, Ontario, Canada, 3 World Child Cancer, London, United Kingdom, 4 Faculty of Medicine, Dalhousie University, Halifax, Nova Scotia, Canada, 5 Cambridge Palliative and End of Life Care Group, Primary Care Unit, Department of Public Health and Primary Care, University of Cambridge, Cambridge, United Kingdom, 6 Faculty of Medicine, Queens University, Kingston, Ontario, Canada, 7 MWAPO Health Development Group, Nairobi, Kenya, 8 Centre for Humanitarian Leadership, Faculty of Arts and Education, Deakin University, Burwood, Victoria, Australia, 9 Usher Institute of Population Health Sciences and Informatics, Global Health Academy, Centre for Population Health Sciences, University of Edinburgh, Edinburgh, United Kingdom, 10 Health Protection and Emergency Management, Department of Health and Human Services, Melbourne, Victoria, Australia, 11 Fasiuddin Khan Research Foundation, Dhaka, Bangladesh

* mdoherty@cheo.on.ca

**Data Availability Statement:** Data cannot be shared publicly due to privacy concerns. Data are available from the Children's Hospital of Eastern

## Abstract

### Background

Despite recognition that palliative care is an essential component of any humanitarian response, serious illness-related suffering continues to be pervasive in these settings. There is very limited evidence about the need for palliative care and symptom relief to guide the implementation of programs to alleviate the burden of serious illness-related suffering in these settings. A basic package of essential medications and supplies can provide pain relief and palliative care; however, the practical availability of these items has not been assessed. This study aimed to describe the illness-related suffering and need for palliative care in Rohingya refugees and caregivers in Bangladesh.

### Methods and findings

Between November 20 and 24, 2017, we conducted a cross-sectional study of individuals with serious health problems (*n* = 156, 53% male) and caregivers (*n* = 155, 69% female) living in Rohingya refugee camps in Bangladesh, using convenience sampling to recruit participants at the community level (i.e., going house to house to identify eligible individuals). The serious health problems, recent healthcare experiences, need for medications and medical supplies, and basic needs of participants were explored through interviews with trained Rohingya community members, using an interview guide that had been piloted with Rohingya individuals to ensure it reflected the specificities of their refugee experience and culture.

Ontario Research Ethics Board (contact via Brooklyn Ward, bward@cheo.on.ca) for researchers who meet the criteria for access to confidential data.

**Funding:** World Child Cancer provided unrestricted funding to MD for completion of the research project (no grant number) (https://www.worldchildcancer.org). The funders had no role in study design, data collection and analysis, decision to publish, or preparation of the manuscript.

**Competing interests:** The authors have declared that no competing interests exist.

**Abbreviations:** LMICs, low-and middle-income countries; NGO, non-governmental organization; TB, tuberculosis.

The most common diagnoses were significant physical disabilities ($n = 100$, 64.1%), treatment-resistant tuberculosis (TB) ($n = 32$, 20.5%), cancer ($n = 15$, 9.6%), and HIV infection ($n = 3$, 1.9%). Many individuals with serious health problems were experiencing significant pain (62%, $n = 96$), and pain treatments were largely ineffective (70%, $n = 58$). The average age was 44.8 years (range 2–100 years) for those with serious health problems and 34.9 years (range 8–75 years) for caregivers. Caregivers reported providing an average of 13.8 hours of care per day. Sleep difficulties (87.1%, $n = 108$), lack of appetite (58.1%, $n = 72$), and lack of pleasure in life (53.2%, $n = 66$) were the most commonly reported problems related to the caregiving role. The main limitations of this study were the use of convenience sampling and closed-ended interview questioning.

## Conclusions

In this study we found that many individuals with serious health problems experienced significant physical, emotional, and social suffering due to a lack of access to pain and symptom relief and other essential components of palliative care. Humanitarian responses should develop and incorporate palliative care and symptom relief strategies that address the needs of all people with serious illness-related suffering and their caregivers.

---

### Author summary

#### Why was this study done?

- Palliative care and symptom relief have been recognized as essential in humanitarian crises, which by their nature generate a large burden of suffering and mortality.

- A basic and inexpensive package of essential medicines and supplies can address pain relief and palliative care during humanitarian crises, but the availability of these items during a crisis has not been assessed.

- There is minimal evidence to guide the implementation of palliative care in humanitarian responses, with few studies describing palliative care needs or programs in these settings.

#### What did the researchers do and find?

- We conducted a cross-sectional study of individuals with serious illnesses and caregivers to describe the illness-related suffering and need for palliative care in Rohingya refugees in Bangladesh.

- Many individuals with serious health problems experienced significant pain (62%, $n = 96$), the pain treatments prescribed were largely ineffective (70%, $n = 58$), and effective pain treatments were rarely available.

- Caregivers most commonly assisted with bathing ($n = 117$, 94%), administering medications ($n = 99$, 80%), and feeding ($n = 98$, 79%).

- Despite having limited training or money, caregivers provided many hours per day of care, which caused sadness, worry, and discrimination.

## What do these findings mean?

- In the Rohingya humanitarian crisis, the specific physical, emotional, and social needs of individuals with serious conditions and their caregivers are not being addressed adequately.

- Efforts to incorporate palliative care must address the barriers to accessing essential medications, supplies, and medical care, including opioid availability.

- Assessments of palliative care needs during humanitarian crises should be used to direct palliative care priorities and guide the development of effective interventions in these settings.

- Future research should quantify the occurrence of serious illness-related suffering, evaluate programs designed to alleviate this suffering, and validate published guidelines and recommendations.

## Introduction

Humanitarian crises, by their nature, generate a large burden of suffering and mortality, necessitating palliative care [1]. A recent Lancet Commission report on pain relief and palliative care recognized palliative care as "an essential component of any response to humanitarian emergencies and crises" [2]. Despite the growing recognition of the need for palliative care in humanitarian settings, its provision has largely been neglected, due to a focus on saving lives [3]. In humanitarian settings, the need for palliative care and symptom relief often extends beyond individuals with life-limiting conditions. Illness-related suffering may occur for many individuals with serious acute or non-life-threatening conditions due to limited access to services to prevent, diagnose, or treat disease and limited social support systems [2]. The role of palliative care in a humanitarian crisis should be to respond to the specific needs of the populations experiencing the crisis.

Minimal evidence exists to guide humanitarian organizations in the design, development, and implementation of palliative care services [4,5]. A 2017 systematic review of humanitarian health programs that included palliative care or enhanced pain management identified only one publication fulfilling the selection criteria: a study of a pain treatment program for amputees [6]. Although palliative care projects are taking place in several other humanitarian situations, these have been accompanied by very little formal research [7]. An "essential package" of inexpensive and relatively simple interventions that can deliver effective palliative care and alleviate serious illness-related suffering in a variety of settings has been proposed [2]. Serious illness-related suffering is defined as suffering associated with an illness or injury that "compromises physical, social and emotional functioning" and requires medical intervention to be relieved [2]. To address the significant burden of serious illness-related suffering in humanitarian settings, evidence on palliative care priorities and effective interventions is urgently needed.

Ensuring patients can obtain strong pain medications is essential to reducing serious illness-related suffering, yet in humanitarian settings, this may pose significant challenges since

national regulatory barriers can restrict the importation of opioids with humanitarian emergency supplies [8]. Many countries have opioid regulations that focus on reducing the risks of nonmedical use, but fail to ensure appropriate access for medical needs [8,9]. The World Health Organization (WHO) has published guidelines providing practical instructions on improving opioid availability while ensuring safe storage and dispensing [10]. International humanitarian organizations are beginning to acknowledge the need for improved pain management in the emergency setting, but there are very few examples reported in the literature of efforts to incorporate key policy lessons [1].

This study aims to describe the need for palliative care and symptom relief during an unfolding humanitarian crisis: the Rohingya refugee crisis in Cox's Bazar, Bangladesh. Specifically, we sought to describe the burden of serious illness-related suffering, focusing on physical, social, and emotional suffering, and the availability of the essential package components.

## Methods

The study was approved by the Civil Surgeon for Cox's Bazar District and by the Research Ethics Board of the Children's Hospital of Eastern Ontario, Canada (Study number: 18/54X). The STROBE statement for our paper can be found in S1 STROBE Checklist. Written informed consent was obtained from all participants. All analyses were non-prespecified. The original funding proposal can be found in S1 Text.

### Recruitment and sampling

Individuals with serious health problems and caregivers for such individuals were invited to participate. In determining which patients (i.e., individuals with serious health problems) to include, we used WHO guidelines for palliative care in humanitarian settings, which suggest that palliative care is appropriate for those with serious health problems, including life-limiting conditions as well as non-life-limiting conditions, such as trauma, burns, paraplegia, quadriplegia, brain injuries, and congenital anomalies where significant suffering may occur [4]. Participants were identified at the community level. Interviewers spoke to individuals in the refugee camp community and went from house to house to identify individuals requiring palliative care or symptom relief and their caregivers. Once identified, these individuals were approached about their interest to learn about the study; if individuals expressed interest, then the interviewer explained the goals and process of the study, responded to any questions, and then asked if the individual would be willing to participate. For individuals whose diagnosis was uncertain, interviewers reviewed the cases with the study coordinators (MD and FK) to determine if the individual should be included as having a serious health problem, prior to conducting the interview. Patients who had significant impairments in movement, muscle tone, and/or balance were categorized as having a significant physical disability, and the breadth of this category was due to a lack of clarity from individuals about their exact diagnosis or the cause of their disability and a lack of access to diagnostic healthcare services. For children (0–17 years), a parent or the primary adult caregiver was approached for consent and completed the interview as a source of proxy information. Participants were identified through convenience sampling, and sample size was determined by the maximum number of eligible participants that could be consented and interviewed during the data collection period. For all pharmacies that could be identified in the defined locations for the study, we interviewed a pharmacy representative about the availability of essential palliative care medications and supplies.

## Design and content of interviews

The interview guide was developed through a literature review that identified key themes from previous assessments of palliative care in low-and middle-income countries (LMICs) and from the 2017 draft Sphere Handbook [5,11,12]. Six of the study authors (MD, MP, RP, LG, BS, and FK) and 3 additional individuals with expertise in humanitarian medicine, palliative care, and noncommunicable diseases provided feedback on the validity and comprehensibility of draft interview questions, which led to the development of a pilot interview guide. This pilot guide was tested with 10 Rohingya interviewers from the refugee camps and 20 Rohingya individuals (11 with serious health problems and 9 caregivers). Modifications were made to improve the clarity of questions and response options to reflect the specificities of the Rohingya refugees' experiences and culture. The pharmacy representative interview included questions about medications and supply items included in the essential package [2].

Demographic information about individuals' age, sex, household size, education, and occupation was collected. Participants with serious health problems were asked to report on the characteristics of their pain and other symptoms, including severity, and treatments and their efficacy. Participants with serious health problems and caregivers were also asked about their needs for medications and medical supplies, as well as their basic needs for items such as food, shelter, and money. Participants with serious health problems were asked about recent health-care experiences and the barriers to accessing care, medicines, and medical supplies.

Individuals were informed that they could skip any questions that they did not want to answer. This, as well as the lack of relevance of certain items, led to variable sample size for responses across certain interview items. In all such cases, the sample size is noted in the text and/or tables.

## Interviewer training and linguistic adaptation

Two co-authors (MD and FK), who had previous experience with conducting similar studies in Bangladesh, recruited and trained 10 Rohingya-speaking interviewers to conduct structured interviews for this study. The interviewers were identified through a partnership with a local health non-governmental organization (NGO) working in the refugee camps (OBAT Helpers) that had previously employed the majority of these individuals in various health promotion and/or translator roles for programs in the refugee camps. All interviewers had completed secondary school and were fluent in both written and spoken English and in the Rohingya language (spoken only, as there is no widely accepted written form of Rohingya). All interviewers were of Rohingya ethnicity, and the majority (90%) had been living in the refugee camps for more than 5 years, having arrived during previous waves of refugee movement from Myanmar.

Interviewer training was conducted in English and consisted of 2 days of theoretical and practical in-person training that included didactic teaching and practical examples related to research ethics and informed consent, the goals of the study, and key concepts related to the study (palliative care, serious health problems, cancer, HIV/AIDS, medications, medical equipment, and symptoms). During the training, the interviewers reviewed each question in the interview guide with the trainers, discussed the meaning of the question and the response options, and provided suggestions about cultural or other adaptations that would improve the clarity of the interview guide. The interviewers then agreed upon a single translation of each interview question into Rohingya, after group discussion. During the practical portion of the training, interviewers conducted practice interviews in pairs, with observation and coaching by the trainers, who provided feedback about interview technique and clarified the interview guide questions and response options. During the final phase of training, interviewers

conducted interviews in the refugee camps, under the same conditions as those in the actual study, with observation by the 2 trainers, to ensure fidelity to the interview guide. The interviews were conducted over a 5-day period (November 20–24, 2017) immediately following training. Interviews typically took 30–45 minutes.

## Setting

Between August and November 2017, violence towards Rohingya people in Myanmar forced 687,000 people into Bangladesh, where an estimated 213,000 Rohingya refugees were already living [13]. Since arriving, the majority are living in makeshift accommodations, and it is estimated that 55% of the newly arrived are children [13]. Interviews with patients and caregivers were conducted among Rohingya refugees living in the main refugee settlement areas of Kutupalong, Jamtoli, Tenkhali, and Balukhali in the Cox's Bazar District of Bangladesh between November 20 and 24, 2017. Retail pharmacy representatives in the refugee camp area, the nearest town center (5.5 km from the entrance to Kutupalong refugee camp), and the nearest government health complex (7.0 km from the Kutupalong camp entrance) were also interviewed. There are many NGO medical clinics (with basic outpatient facilities) and hospitals (inpatient and outpatient facilities) located within the Rohingya settlements or in close proximity, where basic and advanced-level medical and surgical care is available. Additionally, refugees may visit Bangladesh government health facilities, with the nearest primary-level government facility 7.0 km from the Kutupalong camp and a tertiary facility 37 km away.

## Data analysis

Descriptive statistics were obtained using Microsoft Excel.

## Results

### Sample size and composition

There were 311 individuals who participated in this study, including 156 persons living with serious health problems and 155 current or bereaved caregivers. The majority (198, 70.7%) of participants had arrived in Bangladesh in the past 6 months, and nearly all individuals had arrived within the last year ($n = 247$, 88.2%).

### Socio-demographic profile: Individuals with serious health problems

The mean age of those with serious health problems was 44.8 years (median 42, range 2–100 years), and 52.6% ($n = 82$) were male. The majority ($n = 141$, 90.4%) reported having no formal schooling beyond primary level. The most common diagnoses reported were significant physical disabilities ($n = 100$, 64.1%), treatment-resistant tuberculosis (TB) ($n = 32$, 20.5%), cancer ($n = 15$, 9.6%), and HIV infection ($n = 3$, 1.9%). For patients with HIV infection, cancer, or TB, 55.8% ($n = 29$) had received disease-directed treatment (i.e. anti-retroviral therapy, chemotherapy, or TB treatment) while in Myanmar, and 23.1% ($n = 12$) continued to receive these treatments upon arrival in Bangladesh. Additional demographic and diagnostic data are shown in Table 1.

### Physical symptoms and greatest needs

Pain due to the serious illness in the past 3 days was reported by 110 patients (70.5%) and was frequently self-rated as being of moderate ($n = 48$, 30.8%) or severe intensity ($n = 48$, 30.8%). There were 83 patients (53.2%) who reported receiving medication for their pain. Nearly half ($n = 51$, 46.4%) of those who received treatment for pain could not recall the name of the

**Table 1. Socio-demographic data for individuals with serious health problems (*n* = 156).**

| Characteristic | Prior to leaving Myanmar | | Current | |
|---|---|---|---|---|
| | *n* | Percent or mean; range (SD) | *n* | Percent or mean; range (SD) |
| **Sex** | | | | |
| Male | | | 82 | 52.6% |
| Female | | | 73 | 46.8% |
| Missing data | | | 1 | 0.6% |
| **Age (years)** | | | | |
| 0–4 | | | 8 | 5.1% |
| 5–17 | | | 15 | 9.6% |
| 18–49 | | | 58 | 37.2% |
| ≥50 | | | 68 | 43.6% |
| Missing data | | | 7 | 4.5% |
| **Highest level of education completed** | | | | |
| None | | | 87 | 55.8% |
| Primary | | | 54 | 34.6% |
| Secondary | | | 11 | 7.1% |
| Post-secondary (college) | | | 2 | 1.3% |
| Missing data | | | 2 | 1.3% |
| **Primary serious health problem** | | | | |
| Significant physical disability[a] | | | 100 | 64.1% |
| Treatment-resistant tuberculosis | | | 32 | 20.5% |
| Cancer | | | 15 | 9.6% |
| HIV infection | | | 3 | 1.9% |
| Burns | | | 2 | 1.3% |
| Diabetes | | | 1 | 0.6% |
| Chronic respiratory disease | | | 1 | 0.6% |
| Kidney disease | | | 1 | 0.6% |
| Intellectual disability | | | 1 | 0.6% |
| **Number of individuals living in household[b]** | | | 150 | 5.3; 1–19 (2.4) |
| **Number of children (0–18 years of age) the individual has[c]** | | | 109 | 1.7; 0–8 (1.9) |
| **Occupation** | | | | |
| Homemaker | 40 | 25.6% | 21 | 13.5% |
| Farmer | 34 | 21.8% | 2 | 1.3% |
| Unemployed | 31 | 19.9% | 70 | 44.9% |
| Self-employed | 7 | 4.5% | 0 | 0.0% |
| Missing data | 7 | 4.5% | 7 | 4.5% |
| Other[d] | 5 | 3.2% | 2 | 1.3% |
| Public sector worker | 2 | 1.3% | 0 | 0.0% |
| Unable to work due to illness | 4 | 2.6% | 29 | 18.6% |
| Teacher | 2 | 1.3% | 2 | 1.3% |
| Student | 1 | 0.6% | 0 | 0.0% |
| Child <18 years old (therefore no occupation) | 23 | 14.7% | 23 | 14.7% |

[a]Includes individuals with cerebral vascular accident, spinal cord injury, cerebral palsy, and undiagnosed conditions resulting in similar impairments.

[b]Missing data for 6 patients.

[c]Includes individuals ≥18 years of age, and individuals <18 years of age if they are married.

[d]Includes carpenter, tailor, shopkeeper, and unspecified responses.

medication. The most frequent medications reported were paracetamol ($n$ = 19, 17.3%) and non-steroidal anti-inflammatories ($n$ = 9, 8.2%). Only 1 patient (0.9%) reported receiving an opioid pain medication, oxycodone. In addition to pain, individuals with serious illnesses reported an average of 3.8 (SD 2.2, range 0–9) other symptoms that were causing them physical discomfort. When asked about their greatest needs, patients most frequently reported medications (97.4%, $n$ = 152), money (94.2%, $n$ = 147), and food (76.9%, $n$ = 120). Table 2 shows additional details of physical symptoms and greatest needs reported by patients.

**Table 2. Pain severity, treatment, treatment outcomes, and greatest needs ($n$ = 156).**

| Item | $n$ | Percent |
|---|---|---|
| **Severity of pain caused by illness** | | |
| None | 41 | 26.3% |
| Mild | 14 | 9.0% |
| Moderate | 48 | 30.8% |
| Severe | 48 | 30.8% |
| Missing data | 5 | 3.2% |
| **Treatments received by individuals experiencing pain[a] ($n$ = 110) (medications identified in bold are potentially pain relieving)** | | |
| **Paracetamol** | 19 | 17.3% |
| **Non-steroidal anti-inflammatory drug** | 9 | 8.2% |
| **Gastro-esophageal reflux treatment** | 9 | 8.2% |
| Antibiotic | 5 | 4.5% |
| Vitamins/minerals | 4 | 3.6% |
| Antihistamine | 4 | 3.6% |
| **Gabapentinoid** | 2 | 1.8% |
| Other prescription medication | 2 | 1.8% |
| **Oxycodone** | 1 | 0.9% |
| Cannot identify or remember name of treatment | 51 | 46.4% |
| Missing data | 4 | 3.6% |
| Did not receive treatment for pain | 23 | 20.9% |
| **Pain severity after treatment ($n$ = 83)** | | |
| None | 3 | 3.6% |
| Mild | 22 | 26.5% |
| Moderate | 31 | 37.3% |
| Severe | 27 | 32.5% |
| **Other physical symptoms[a]** | | |
| Fever | 116 | 74.4% |
| Sleep difficulties | 86 | 55.1% |
| Cough | 81 | 51.9% |
| Lack of appetite | 71 | 45.5% |
| Breathing problems | 66 | 42.3% |
| Fatigue | 51 | 32.7% |
| Nausea/vomiting | 39 | 25.0% |
| Diarrhea | 17 | 10.9% |
| Other[b] | 29 | 18.6% |
| **Ill individuals' greatest needs[a]** | | |
| Medications | 152 | 97.4% |
| Money | 147 | 94.2% |
| Food | 120 | 76.9% |

*(Continued)*

**Table 2.** (Continued)

| Item | n | Percent |
|---|---|---|
| Pain relief | 72 | 46.2% |
| Someone to help | 70 | 44.9% |
| Love | 47 | 30.1% |
| Help with sadness or depression | 40 | 25.6% |
| Respect | 35 | 22.4% |
| Care for my children | 32 | 20.5% |
| Schooling for my children | 24 | 15.4% |
| A job or source of income | 10 | 6.4% |
| Medical equipment | 4 | 2.6% |
| Other[c] | 4 | 2.6% |

[a]Patients could provide more than 1 response.

[b]Includes weight loss, bone/joint pains, anxiety, paralysis, loss of sensation, headache, drooling, constipation, muscle spasms, bleeding, swelling, dizziness, coryza.

[c]Includes sympathy or understanding, medical imaging, a car, and a toilet.

## Essential medications and medical equipment

Sixty-one patients (39.1%) reported needing medications, including paracetamol ($n = 21$, 34.4%), antibiotics ($n = 16$, 26.2%), medications for chronic diseases ($n = 16$, 26.2%), and medications for gastro-esophageal reflux ($n = 14$, 23.0%). Only 52.5% ($n = 32$) of these patients were able to access their medications at the time of the interview. Eighty-two patients (52.6%) reported needing at least 1 medical supply item, and 72.0% ($n = 59$) of these patients were unable to access needed equipment. The most commonly needed items were urinary catheters ($n = 29$, 35.4%), adult diapers ($n = 21$, 25.6%), and oxygen ($n = 21$, 25.6%). Table 3 shows the complete list of reported medication and medical supply needs.

**Table 3. Current medication and equipment requirements and barriers to access[a].**

| Item | n | Percent |
|---|---|---|
| **Required medication ($n = 61$)** | | |
| Paracetamol | 21 | 34.4% |
| Antibiotic | 16 | 26.2% |
| Other chronic disease medication[b] | 16 | 26.2% |
| Gastro-esophageal reflux treatment[c] | 14 | 23.0% |
| Vitamins/minerals/nutritional supplements | 13 | 21.3% |
| Medication could not be identified[d] | 12 | 19.7% |
| TB treatment | 8 | 13.1% |
| Non-steroidal anti-inflammatory drug | 6 | 9.8% |
| Medication for minor ailment (includes cough syrups and decongestants) | 6 | 9.8% |
| Antihistamine | 4 | 6.6% |
| Antifungal medication | 4 | 6.6% |
| Gabapentinoid | 3 | 4.9% |
| Topical preparation | 3 | 4.9% |
| Baclofen | 2 | 3.3% |
| Oral rehydration solution | 2 | 3.3% |
| Steroid | 2 | 3.3% |

(*Continued*)

**Table 3.** (Continued)

| Item | n | Percent |
|---|---|---|
| Opioid | 1 | 1.6% |
| Anticonvulsant | 1 | 1.6% |
| Eye drops | 1 | 1.6% |
| **Currently required medical supplies and equipment (n = 82)** | | |
| Urinary catheter | 29 | 35.4% |
| Adult diapers | 21 | 25.6% |
| Oxygen | 21 | 25.6% |
| Wheelchair | 16 | 19.5% |
| Pressure-reducing mattress | 13 | 15.9% |
| Feeding tube (nasogastric tube) | 10 | 12.2% |
| Equipment to assist with using washroom (e.g., commode, bedpan) | 9 | 11.0% |
| Other | 7 | 8.5% |
| **Reported barriers and challenges to access (n = 108)** | | |
| Lack of money | 65 | 60.2% |
| Surrounding health facilities do not have access to the treatment needed | 14 | 13.0% |
| Difficulty getting to healthcare facility, due to distance and/or mobility problems | 11 | 10.2% |
| Individual does not know where to get treatment | 8 | 7.4% |
| Individual believes there is no treatment for their condition | 8 | 7.4% |
| Poor quality of treatment given at health facility | 6 | 5.7% |
| Individual is waiting for TB test results | 1 | 0.9% |
| Individual does not know what treatment he/she needs | 1 | 0.9% |

[a]Individuals could provide more than 1 response.

[b]Includes antihypertensives, salbutamol, theophylline, montelukast, dyslipidemic agents, oxybutynin, antiplatelet agents, and allopurinol.

[c]Includes proton pump inhibitors, H2 blockers, and antacids.

[d]Patient could not recall the name of the medication, or research team could not identify the medication from the response provided.

TB, tuberculosis.

## Patterns of care and challenges

Sixty percent of individuals (n = 93, 59.6%) reported visiting a health facility in the past month. Of those patients, 53.7% (n = 50) reported visiting an NGO hospital, 18.3% (n = 17) a health clinic, and 9.7% (n = 9) a government primary-level health facility. Forty-three percent (n = 40) of patients who visited a healthcare facility sought medical care for concerns related to their serious health problem, and 29.0% (n = 27) visited specifically for pain relief. More than half (59.1%, n = 55) of those who visited a healthcare facility reported that their visit was unsuccessful at treating their presenting problem. Commonly identified barriers and challenges to accessing healthcare included lack of money (n = 65, 60.2%), lack of treatment availability at the health facility (n = 14, 13.0%), and difficulty getting to the healthcare facility (n = 11, 10.2%). Further details of barriers and challenges to healthcare access are found in Table 3.

## Socio-demographic profile: Caregivers

There were 155 caregivers who were interviewed. We excluded from further analysis 31 caregivers who were bereaved longer than 6 months, as their caregiving occurred primarily in Myanmar, prior to the refugee crisis. For the remaining 124 caregivers, the mean age was 34.9

years (median 32 years, range 8–75 years). Caregivers were frequently women ($n = 85$, 68.5%) and were caring for an average of 3.8 children (0–18 years) of their own (range 0–12, SD 2.5). Caregivers were most commonly members of the ill patient's family ($n = 118$, 95.2%) and provided an average of 13.8 hours of care per day (range 2–24, SD 9.4). Further characteristics of caregivers are shown in Table 4.

## Dimensions of caregiving

The most frequent activities performed by caregivers included bathing ($n = 117$, 94.4%), administering medications ($n = 99$, 79.8%), and feeding the ill individual ($n = 98$, 79.0%). Only 11.3% ($n = 14$) reported having received training on how to provide care. Caregivers commonly reported having sleep difficulties (87.1%, $n = 108$), lack of appetite (58.1%, $n = 72$), and lack of pleasure in life (53.2%, $n = 66$) due to their caregiving role. When asked about their greatest needs, caregivers commonly reported money (88.7%, $n = 110$), food (74.2%, $n = 92$), and someone to help them (65.3%, $n = 81$). Further details of the roles, challenges, and needs of caregivers are shown in Table 5.

**Table 4. Socio-demographic data for caregivers ($n = 124$).**

| Characteristic | Prior to leaving Myanmar | | Current | |
|---|---|---|---|---|
| | $n$ | Percent | $n$ | Percent or mean; range (SD) |
| **Sex** | | | | |
| Female | | | 85 | 68.5% |
| Male | | | 38 | 30.6% |
| Missing data | | | 1 | 0.8% |
| **Highest level of education completed** | | | | |
| None | | | 48 | 38.7% |
| Primary | | | 65 | 52.4% |
| Secondary | | | 8 | 6.5% |
| College | | | 2 | 1.6% |
| Missing data | | | 1 | 0.8% |
| **Age (years)** | | | | |
| 8–17 | | | 4 | 3.2% |
| 18–49 | | | 99 | 79.8% |
| ≥50 | | | 16 | 12.9% |
| Missing data | | | 5 | 4.0% |
| **Household size (number of individuals)** | | | 122 | 6; 2–19 (2.5) |
| **Occupation** | | | | |
| Homemaker | 98 | 79.0% | 84 | 53.8% |
| Farmer | 11 | 8.9% | 3 | 1.9% |
| Student | 5 | 4.0% | 2 | 1.3% |
| Other[a] | 3 | 2.4% | 2 | 1.3% |
| Self-employed | 3 | 2.4% | 3 | 1.9% |
| Unemployed | 2 | 1.6% | 25 | 16.0% |
| Public sector worker | 1 | 0.8% | 0 | 0.0% |
| Private sector worker | 1 | 0.8% | 3 | 1.9% |
| Missing data | 0 | 0.0% | 2 | 1.3% |

[a]Includes committee member (unspecified), teacher, fisherman, and unspecified responses.

**Table 5. The roles, challenges, and needs of caregivers (*n* = 124)[a].**

| Item | *n* | Percent |
|---|---|---|
| **Activities performed by caregiver for ill individual** | | |
| Bathing | 117 | 94.4% |
| Administering medications | 99 | 79.8% |
| Feeding | 98 | 79.0% |
| Massage | 68 | 54.8% |
| Providing emotional support | 31 | 25.0% |
| Providing care for pain or other symptoms | 17 | 13.7% |
| Other | 7 | 5.6% |
| **Challenges faced by caregivers** | | |
| Insufficient financial resources | 120 | 96.8% |
| Caregiving is very hard work | 84 | 67.7% |
| Lack of help in caregiving | 58 | 46.8% |
| Feeling sadness | 48 | 38.7% |
| Worrying about the future | 45 | 36.3% |
| Discrimination | 24 | 19.4% |
| Insufficient time | 16 | 12.9% |
| Unsure of how to provide care to ill individual | 18 | 14.5% |
| Other | 1 | 0.8% |
| **Problems experienced by caregivers** | | |
| Difficulty sleeping | 108 | 87.1% |
| Lack of appetite | 72 | 58.1% |
| Lack of pleasure | 66 | 53.2% |
| Stress or anxiety | 46 | 37.1% |
| Not wanting to be with others | 46 | 37.1% |
| Difficulty concentrating | 40 | 32.3% |
| Not feeling anything emotionally | 28 | 22.6% |
| Other | 3 | 2.4% |
| **Caregivers' greatest needs** | | |
| Money | 110 | 88.7% |
| Food | 92 | 74.2% |
| Someone to help me | 81 | 65.3% |
| Love | 32 | 25.8% |
| A way to make money | 30 | 24.2% |
| Help dealing with emotions | 26 | 21.0% |
| Respect | 25 | 20.2% |
| Care for my children | 23 | 18.5% |
| Schooling for my children | 19 | 15.3% |

[a]Participants could provide multiple responses for each question.

## Availability of essential medicines and supplies

Shopkeepers at 17 pharmacies were interviewed about the availability of essential palliative care medications and supplies. Morphine was not available in any pharmacy, and only 1 pharmacy (5.9%) had any suitable oral opioids available, in the form of oxymorphone tablets. Table 6 provides further details of medication and supply availability.

**Table 6. Medicine and medical equipment availability in local pharmacies by location.**

| Item | Number of pharmacies with medication or equipment in stock | | | | |
|---|---|---|---|---|---|
| | Pharmacies in refugee camps (*n* = 7) | Pharmacies in market in nearest town (*n* = 7) | Pharmacies at nearest government primary health center (*n* = 3) | Total number of pharmacies with item available | Percent of all pharmacies (*n* = 17) with item available |
| **Medications from essential package** | | | | | |
| Amitriptyline | 0 | 4 | 2 | 6 | 35.3% |
| Bisacodyl | 0 | 2 | 0 | 2 | 11.8% |
| Dexamethasone | 4 | 6 | 3 | 13 | 76.5% |
| Diazepam | 1 | 6 | 2 | 9 | 52.9% |
| Fluconazole | 4 | 6 | 3 | 13 | 76.5% |
| Furosemide | 2 | 4 | 1 | 7 | 41.2% |
| Fluoxetine or sertraline or citalopram | 0 | 3 | 0 | 3 | 17.6% |
| Hyoscine butylbromide | 1 | 2 | 1 | 4 | 23.5% |
| Haloperidol | 0 | 3 | 0 | 3 | 17.6% |
| Ibuprofen | 1 | 6 | 2 | 9 | 52.9% |
| Lactulose | 3 | 7 | 3 | 13 | 76.5% |
| Loperamide | 3 | 5 | 3 | 11 | 64.7% |
| Metoclopramide | 0 | 0 | 0 | 0 | 0.0% |
| Metronidazole | 4 | 7 | 3 | 14 | 82.4% |
| Morphine—immediate release tablet | 0 | 0 | 0 | 0 | 0.0% |
| Morphine—sustained release tablet | 0 | 0 | 0 | 0 | 0.0% |
| Morphine—injectable | 0 | 0 | 0 | 0 | 0.0% |
| Naloxone—injectable | 0 | 0 | 0 | 0 | 0.0% |
| Ondansetron | 1 | 7 | 3 | 11 | 64.7% |
| Paracetamol | 7 | 7 | 3 | 17 | 100.0% |
| Petroleum jelly | 0 | 1 | 2 | 3 | 17.6% |
| **Medical equipment from essential package** | | | | | |
| Adult diapers | 0 | 0 | 0 | 0 | 0.0% |
| Nasogastric tube | 0 | 1 | 2 | 3 | 17.6% |
| Urinary catheter | 0 | 1 | 2 | 3 | 17.6% |
| **Other opioids available in Bangladesh** | | | | | |
| Fentanyl injection | 0 | 0 | 0 | 0 | 0.0% |
| Nalbuphine injection | 0 | 1 | 2 | 3 | 17.6% |
| Oxymorphone tablets | 0 | 1 | 0 | 1 | 5.9% |
| Pethidine injection | 0 | 0 | 0 | 0 | 0.0% |
| Tramadol tablets | 0 | 5 | 2 | 7 | 41.2% |
| Tramadol suppositories | 0 | 3 | 3 | 6 | 35.3% |
| Tramadol injection | 0 | 2 | 3 | 5 | 29.4% |

## Discussion

We describe the need for palliative care and symptom relief during an unfolding humanitarian crisis, including the illness-related suffering experienced by individuals with serious health problems and the impact on their caregivers of providing care for such individuals. We found that the majority of ill individuals were experiencing significant pain and other physical symptoms and were unable to access the medical treatments necessary to relieve their suffering.

Caregivers provided a considerable amount of care for ill individuals, with significant negative consequences for their own physical and emotional health.

## Pain

**Prevalence and treatment.** A significant proportion of patients (70%) reported pain, and many (21%) had received no medications to treat their pain. To our knowledge, there are no other comparable studies from humanitarian settings, but several studies from India and Bangladesh report similar pain prevalence rates of 71%–100% in patients with serious health problems at the time of initial assessment [14–16]. In our study, those who received treatment frequently reported limited improvement, with 70% reporting moderate or severe pain despite treatment. This may be because the most common pain treatments were paracetamol and non-steroidal anti-inflammatories, which are recommended only for mild pain [17]. Additionally, many of the reported treatments for pain were not analgesics. Our findings support the recent Lancet Commission conclusion that pain contributes significantly to the burden of serious illness-related suffering worldwide [2]. A recently published WHO guide provides practical guidance for humanitarian health actors to implement emergency health system responses that integrate pain and symptom management [4].

**Opioid policy barriers.** We found very little use or pharmacy availability of oral morphine, which is widely accepted as essential for achieving adequate pain control in humanitarian settings [4,18]. Despite morphine and other opioids being included on the Bangladesh Essential Drug List, morphine is not practically available outside of the capital city of Dhaka [14]. The International Narcotics Control Board reports that only 18 kg of morphine was consumed in Bangladesh in 2017, which previous studies have estimated represents less than 1% of the anticipated national opioid need [8,19]. Overly restrictive opioid policies are a common barrier to opioid availability in LMICs; however, several LMICs, including Uganda and Mongolia, provide examples of an appropriately balanced approach to opioid control, which provides for medical needs while addressing the risk of nonmedical use [18,20,21]. Applying these lessons to humanitarian situations may improve morphine availability in these settings. There are no published studies to our knowledge about the availability of opioids in humanitarian crisis situations and the barriers to humanitarian health organizations importing these medications as part of their relief efforts [6]. Humanitarian health organizations may choose not to include opioids in their essential supply packages, fearing that country-specific opioid policies may delay the import of the entire shipment of essential medications and supplies or because of a lack of knowledge among healthcare providers about the safe and effective use of such medications.

**Role of healthcare in pain management.** Many patients had sought healthcare for pain relief, yet few had received adequate pain relief. Barriers to accessing medical care included facilities not having the necessary treatment for the individual's complaint or facilities providing poor quality of treatment. While we did not collect data directly from healthcare professionals, it is plausible that healthcare professionals' fears and misinformation about opioids may have contributed to such patient experiences, since misconceptions about the essential role of opioids in cancer pain relief have been described as a significant barrier to effective pain management in many resource-limited settings [4,18,22]. Education for clinicians about the safe use of opioids can improve pain management for patients, and the Lancet Commission proposed basic mandatory training for all healthcare providers [2,23,24]. Implementation of these strategies should be considered by humanitarian organizations to improve pain management [6].

## Medical interventions

The majority of patients required a range of medical interventions, including medications, medical equipment, visits to health facilities, and basic care. We identified significant barriers to accessing these interventions, including financial problems, unavailable treatments or medications, and a lack of support for caregivers. Humanitarian health facilities generally provide free medicines, but patients are usually given only 1–2 weeks' supply, which creates significant barriers to continuing treatment for individuals with a chronic condition who require medications indefinitely. Despite free medications, the costs for transportation to health facilities and lost wages may be a significant financial burden for patients, and these factors may have contributed to the significant number of patients who reported being unable to obtain necessary medical interventions in our study.

## Caregivers

Family caregivers provided many hours of assistance daily, most often helping with bathing, feeding, and administering medicines, while also providing emotional support. Very few caregivers had received any training in their role. A basic palliative care training manual for caregivers has recently been published, and efforts are underway to adapt this training to humanitarian situations [25]. Previous studies have demonstrated the feasibility of training for family caregivers, showing reductions in caregiver burnout by providing skills to cope with the emotional stress of caregiving [26,27].

## Strengths and limitations

This study, which as far as we are aware is the first formal assessment of palliative care needs in an unfolding humanitarian crisis, used a relatively simple method of assessing palliative care needs, which was easily implemented with limited resources early in a humanitarian crisis. The involvement of local community interviewers enabled the refinement of the study instruments to reflect features of the local culture and is also likely to have facilitated participant recruitment. Despite our efforts to adapt interview items, outstanding issues of cultural relevance and linguistic equivalence may have impacted the assessment. The use of closed-ended interview questions limited the depth and richness of the data, while our sampling methodology may limit the generalizability of the findings.

## Future study

Future studies should validate the methodology we describe, examining its ability to assess palliative care needs and serious illness-related suffering in other populations affected by humanitarian crises or in resource-limited settings. Research priorities should also include quantifying the serious-illness-related suffering experienced by populations in humanitarian settings, evaluating programs designed to alleviate this suffering, and testing published guides and recommendations [2]. Further studies are needed to explore best models for palliative care training in humanitarian situations.

## Supporting information

**S1 STROBE Checklist. STROBE checklist.**
(DOCX)

**S1 Text. Original funding proposal.**
(DOCX)

## Acknowledgments

Mhoira Leng (Makerere University), Joan Marston (PalCHASE), and Jason Nickerson (Bruyère Research Institute and University of Ottawa Centre for Health Law, Policy and Ethics) provided valuable insights into the development of the study instruments and study design.

## Author Contributions

**Conceptualization:** Megan Doherty, Farzana Khan.

**Formal analysis:** Liam Power, Mila Petrova, Scott Gunn, Rachel Coghlan, Liz Grant, Brett Sutton.

**Investigation:** Megan Doherty, Farzana Khan.

**Methodology:** Mila Petrova, Richard Powell.

**Writing – original draft:** Megan Doherty, Liam Power.

**Writing – review & editing:** Megan Doherty, Liam Power, Mila Petrova, Scott Gunn, Richard Powell, Rachel Coghlan, Liz Grant, Brett Sutton, Farzana Khan.

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
