## [Decision Letter · Decision Letter 0]

31 Oct 2019

Dear Dr. Doherty,

Thank you very much for submitting your manuscript "Assessing the serious health-related suffering and need for palliative care in an unfolding humanitarian emergency: A cross-sectional observational study of Rohingya refugees in Bangladesh" (PMEDICINE-D-19-03771) for consideration at PLOS Medicine. 

Your paper was discussed among the editorial team and sent to independent reviewers, including a statistical reviewer. The reviews are appended at the bottom of this email and any accompanying reviewer attachments can be seen via the link below:

[LINK]

In light of these reviews, we will not be able to accept the manuscript for publication in the journal in its current form, but we would like to invite you to submit a revised version that fully addresses the reviewers' and editors' comments. You will appreciate that we cannot make a decision about publication until we have seen the revised manuscript and your response, and we expect to seek re-review by one or more of the reviewers. 

We hope to receive your revised manuscript within two weeks. Please email us (plosmedicine@plos.org) if you have any questions or concerns.

Please let me know if you have any questions. Otherwise, we look forward to receiving your revised manuscript soon. 

Sincerely,

Richard Turner PhD, for Clare Stone, PhD

Managing Editor, PLOS Medicine

rturner@plos.org

Please begin the process of depositing study data in a public repository. 

Please revise your title to better match journal style. We suggest "Illness-related suffering and need for palliative care in Rohingya refugees and caregivers in Bangladesh: a cross-sectional study". 

To your abstract, please add the start and end dates of recruitment for your study; and brief demographic details for study participants. 

The final sentence of the "methods and findings" subsection of your abstract should summarize the study's main limitations. 

Around line 67, please begin your discussion of conclusions by referring to your findings (i.e., "In this study, we found that ..." or similar). 

After the abstract, we will need to ask you to add a new and accessible "author summary" section in non-identical prose. You may find it helpful to consult one or two recent research papers published in PLOS Medicine to get a sense of the preferred style. 

The "Lancet Commission" is mentioned three times in your introduction, and we suggest that once would suffice. 

Early in the methods section, please state whether the study had a protocol or prespecified analysis plan, and if so attach this as a supplementary file (referred to in the text). Please highlight analyses that were not prespecified. 

Throughout your text, please format reference call-outs as follows: "...saving lives [1,3]."

Please avoid using italics for emphasis. 

Please substitute "sex" for "gender" as appropriate. 

Please remove the "Role of the funding source" section from your text. 

In your reference list, please ensure that journal name abbreviations are consistently observed (e.g., "Lancet" for references 2 and 3; "JAMA" for reference 27. 

Please add full access details to references 2 and 7.

Please add author names where available, e.g. to references 11 and 12; please format the author names for reference 1 consistently with other references; and make the author of reference 4 "WHO" (or the fully spelt-out version). 

Please correct the typo in reference 12. 

You mention the STROBE reporting guideline in your text, and we ask you to add a completed checklist as a supplementary file, referred to in the methods section. In the checklist, individual items should be referred to by section (e.g., "Methods") and paragraph number rather than by line or page numbers, as the latter generally change in the event of publication. 

Comments from the reviewers:

*** Reviewer #1: 

The authors have assessed the symptom load and the palliative care needs of a large convenience sample of patients and caregivers from the Rohingya refugee communities in Bangladesh. They found significant physical, emotional and social suffering, lack of access to medicines and other medical supplies, and high burden on caregivers. 

The survey covers a most important topic, which has been neglected by research until now. The authors are to be recommended for the clear structure of their survey and the concise and clear presentation of the results. I have only a few minor comments. 

Reference 4 needs to be corrected (World Health Organization)

Reference 7 does not seem to be complete

Page 7, line 128: When exactly was the data collection period? 

Page 7, line 134: how many experts provided feedback? 

Page 7, line 138: did the authors consider cultural adaptation of the questions and if so, how?

Page 8, line 156: how many interviewers were trained? Did I understand correctly that there was only one training workshop and an 8-day sampling period immediately following that workshop? When did you have the workshop? What contents were included in the practical and the theoretical parts of the workshop?

Page 10, line 188: are you able to estimate how many patients were approached in the recruitment period? Where die the interviewers identify and approach patients? Did they participate in clinics? 

Page 13, table 2: antibiotics or vitamins /minerals would not be considered pain relieving medication. Could you include this point in the discussion?

Page 21, line 333: you should compare the lack of opioid availability and accessability to the INCB data or any other data on opioid consumption in Bangladesh

Page 21, line 339: do humanitarian organizations such as MSF bring opioids to the Rohingya refugee camps? If not, why not? 

Page 23, line 379: another limitation that you should discuss here is that you could not quantify SHS. The Lancet report gave a good methodology for quantification of SHS, and I would have loved to see some data for example on days with SHS. 

*** Reviewer #2: 

I confine my remarks to statistical and methodological aspects of this paper.

I marked "proceed without recommendation" because I can't decide between "reject" and "accept". The statistics themselves are very simple and there's nothing really to do to improve them. But (as the authors acknowledge) this is a convenience sample of a rather difficult population; it's unclear what it says about any other population in any other, similar, situation. (To be fair, the authors don't say that it says anything about such people). 

The problem is not the article itself but what people may make of it. I know (and I am sure the editors do too) that limitations of a study often get ignored. I don't know how to deal with that in this case, hence my "no recommendation". I leave it to the editors to judge. 

Other than that, my only concern is the translation into Rohingya. Translation is tricky; translation of questions about pain can be particularly tricky. I remember one study (although I don't remember the citation) of menstrual pain in bilingual Chinese-American women. The questions were carefully translated, then back tranlsated into English, then edited. The questionnaire was given to the women in both languages and there were significantly different results.\\

Sorry to be so vague but I am not sure what to recommend here.

Peter Flom

*** Reviewer #3: 

1) Congratulations on this important effort to generate evidence of palliative care needs in humanitarian crises.

2) I applaud your inclusion in your study of caregiver burden.

3) The term "health-related suffering" is an oxymoron. While it was used in the Report of the Lancet Commission on Palliative Care, this seems misguided. Please use instead "illness-related suffering."

4) Per the WHO document entitled "Integrating palliative care and symptom relief into the response to humanitarian emergencies and crises: a WHO guide." palliative care is not only for people with "life-limiting condition." In humanitarian crises, many people, perhaps most people, who suffering requires palliative care may not have a "life-limiting condition." Please make clear, per WHO guidance, that palliative care should respond to the local need and that this need varies by location, culture, and socio-economic situation.

5) Your study population is not limited to people with "life-limiting conditions." It includes people with "significant physical disability", burns, drug-resistant TB, and HIV infection. At least some of these people may live long lives, and some may lead relatively healthy long lives. Thus, please change the term "life-limiting condition" to "serious health problem."

6) Line 70: Should state: " ... palliative care which addresses the needs of all people with serious illness-related suffering and their caregivers."

7) Line 77: More relevant and up to date citation: Krakauer EL, Daubman BR, Aloudat T. Integrating palliative care and symptom relief into responses to humanitarian crises. Med J Australia 2019;211;201-203.

8) Lines 107-108: Text should recommend the essential package of palliative care described by WHO specifically for humanitarian crises, not the general package from the Report of the Lancet Commission.

9) In light of the documented lack of access to disease treatment, please emphasize that palliative care is never a substitute for disease treatment and that efforts to make palliative care accessible for people affected by humanitarian crises should be accompanied by and integrated with efforts to make disease prevention, early diagnosis and treatment accessible to the same population.

***

[LINK]

---

## [Decision Letter · Decision Letter 1]

21 Nov 2019

Dear Dr. Doherty,

Thank you very much for re-submitting your manuscript "Illness related-suffering and need for palliative care in Rohingya refugees and caregivers in Bangladesh: a cross-sectional study" (PMEDICINE-D-19-03771R1) for consideration at PLOS Medicine.

I have discussed the paper with editorial colleagues and our academic editor, and it was also seen again by two of our reviewers. I am pleased to tell you that, provided the remaining editorial and production issues are dealt with, we expect to be able to accept the paper for publication in the journal.

[LINK]

We hope to receive your revised manuscript within around one week. Please email us (plosmedicine@plos.org) if you have any questions or concerns.

Please let me know if you have any questions, and otherwise we look forward to receiving your revised manuscript shortly. 

Kind regards,

Richard Turner, PhD

rturner@plos.org

Requests from Editors:

Please finalize the arrangements for data deposition. 

Around lines 57-58 of your abstract, please add an additional 1-2 sentences to describe your approaches. For example, please summarize your methods of recruitment, sampling, health assessment and interviewing. 

Please add summary information on participants' ages to your abstract.

At line 60, please amend the text to "... individuals reported experiencing significant pain" or similar. 

Around line 63 of the abstract, we suggest adding an additional sentence, say, summarizing caregivers' health and wellbeing.

At line 64, we suspect that "closed-ended" should be substituted. 

At line 95, please adapt the text to "... and their caregivers are not being addressed adequately." or similar.

At line 101, please adapt the text to "... should quantify serious illness-relates suffering ...".

We ask you to refer to the attached STROBE checklist in your main text, perhaps around line 150.

You note in your responses that "We have clarified that there was no prespecified analysis plan."; please add a statement to this effect around line 150, including additional wording such as "All analyses were non-prespecified.". 

As an alternative, is there a submission to a funding body, say, that could be uploaded as a supplementary file in place of a study plan?

Please make that "problems" at line 184.

At lines 464-465, we suspect that "... should quantifying the need for serious illness-related suffering" should become "... should quantify serious illness-related suffering" or similar.

Please reformat reference call-outs throughout your text, e.g., "... care services [4,5]."

Please amend the formatting of author names for reference 1, and address any journal names needing abbreviation (e.g., "Lancet" for reference 2). 

You note in the STROBE checklist that information on funding was "removed on the request of the editor". We ask you to substitute "See Metadata" or similar (the current wording could be taken to indicate an intent to conceal, whereas the aim is to present the information in a standardized format rather than in the main text).

Comments from Reviewers:

*** Reviewer #1: 

The authors have revised the paper and fully included all comments from the reviewers. As for the open question on table 2: I like the way they have solved this issue. 

*** Reviewer #3: 

Efforts to make palliative care accessible to the global poor have been sharply and justifiably criticized in the past if they were not linked to efforts to make disease prevention, diagnosis and treatment accessible. If there is a focus purely on relieving symptoms without trying to prevent, diagnose and treat the illnesses that CAUSE the symptoms, this endeavor risks being called immoral and "second rate care for the poor." While I have approved the current draft of the paper, I still think you should consider emphasizing that palliative care in humanitarian crises should be linked to efforts to prevent and treat serious illnesses and to strengthen local healthcare systems.

***

[LINK]

---

## [Editor Report · Decision Letter 2]

24 Jan 2020

Dear Dr Doherty, 

On behalf of my colleagues and the academic editor, Dr. Terry McGovern, I am delighted to inform you that your manuscript entitled "Illness-related suffering and need for palliative care in Rohingya refugees and caregivers in Bangladesh: a cross-sectional study" (PMEDICINE-D-19-03771R2) has been accepted for publication in PLOS Medicine. 

PRODUCTION PROCESS

PRESS

PROFILE INFORMATION

Thank you again for submitting the manuscript to PLOS Medicine. We look forward to publishing it. 

Best wishes, 

Richard Turner, PhD

Senior Editor 

PLOS Medicine

plosmedicine.org